# Juding the Judges: A Systematic Investigation of Position Bias in Pairwise Comparative Assessments by LLMs

## Abstract

LLM-as-a-Judge presents a promising alternative to human evaluators across various tasks, but inherent biases, especially **position bias** — a tendency to favor solutions based on their position in the prompt — have compromised its effectiveness. Our study introduces a systematic framework to examine position bias in pairwise comparisons, focusing on repetition stability, position consistency, and preference fairness. This research significantly contributes to the field by introducing new concepts for understanding position bias and providing a multi-dimensional framework for evaluations. We conducted experiments with 12 LLM judges across MTBench and DevBench, covering 22 tasks and approximately 40 solution-generating models — candidates, resulting in over 100,000 evaluation instances. Our findings confirm that position bias in capable LLM judges is not due to random chances, along with notable variations observed across judges and tasks. Moreover, position bias is weakly influenced by the length of prompt components but significantly impacted by the quality gap between solutions. These insights can help optimize judge model selections, improve benchmark design, and inform future research on debiasing strategies, ultimately enhancing the reliability of LLM judges.

## 1 Introduction

In recent years, Large Language Models (LLMs) have emerged as evolutionary technologies, gathering global interest and stimulating substantial research into their applications. Evaluating large language models (LLMs) has received increasing attention due to their advancing capabilities across diverse fields. While human assessment is considered the gold standard for aligning with human preferences, it lacks scalability and reproducibility in extensive evaluations (Zeng et al., 2023; Karpinska et al., 2021). To automate evaluations and reduce reliance on costly human evaluators, the LLM-as-a-Judge methodology emerged as a promising alternative across various tasks. Despite a high level of agreement with human judgments(Zheng et al., 2024b; Li et al., 2024; Zhu et al., 2023b), inherent biases, especially **position bias**, have undermined the accuracy, fairness, and reliability of these LLM evaluators.

**Position bias** refers to the tendency of LLM judges to favor certain positions within prompt components rather than the content itself, as shown in Fig. 1. This bias is prevalent across all categories of LLM-as-a-Judge, including pointwise, listwise, and pairwise evaluations (Qin et al., 2024), for any score-based and relation-based settings (Li et al., 2023d). Consequently, the reliability of LLM judges becomes questionable when they exhibit position bias. However, addressing this issue is highly complicated due to the varying performance of LLM judges across different tasks (Khan et al., 2024; Chua et al., 2024). Moreover, the limited understanding of the bias itself has exacerbated the problem.

Prior studies on position bias implicitly assume that this bias is not due to random variations. In other words, these studies assume that LLM judges produce consistent results across repeated trials. However, without validating this assumption, evaluations may become less reliable. As shown in Fig. 1, the occurrence of position bias can be observed by chance even for the same set of tasks and candidate models. Thus, it is unclear whether the observed position bias actually arises from the

position of the prompt components or is merely due to random chance. Therefore, when exploring the position bias of LLM judges, it is necessary for the judges to be stable across repetitions to yield nontrivial results.

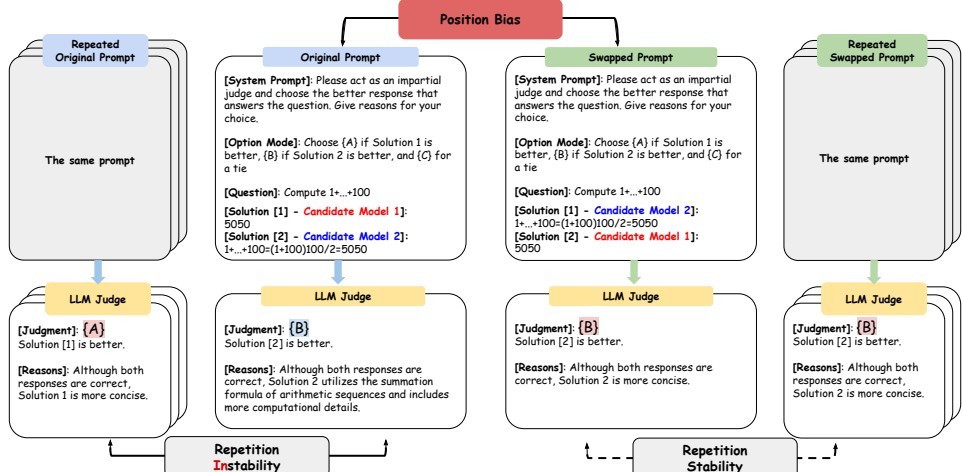

Figure 1: Demonstration of **position bias** and **repetition stability** of LLM-as-a-Judge. When the positions of candidate model solutions are swapped in the prompt, position bias is exhibited if the LLM judge tends to favor the same position (e.g., always choosing {B}) rather than the same candidate model solution. We observe **repetition instability** if the same LLM judge gives different judgments across repetitions. Only when an LLM judge has high repetition stability can we validate that its position bias is not due to random variations.

Another key aspect of position bias, **preference fairness**, has also been under-explored by previous works. Most studies have primarily focused on *position consistency* to measure how frequently position bias occurs across a set of evaluation instances. However, the direction of biased preference (i.e., **primacy** or **recency**) is also insightful. Fig. 2 illustrates cases where a pair of judgments with swapped prompts become position consistent, inconsistent, primacy-preferred, or recency-preferred. We measure **preference fairness** by analyzing the distribution of these preference directions across a set of evaluations. Specifically, judgments are considered to have a fair preference if they are position consistent or if the preference directions are evenly distributed. In contrast, an LLM judge is considered to have an unfair preference if it consistently favors candidate model solution that appear either first or last in the prompt. Both fair and unfair judge models can provide valuable insights into model properties, applications, and strategies for improvement.

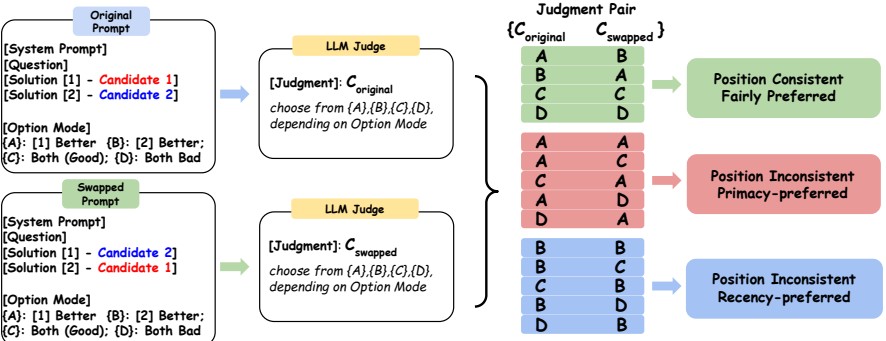

Figure 2: Demonstration of position consistent/inconsistent, primacy-preferred, and recency-preferred judgment pairs in pairwise comparative assessment, where the task is to choose the better candidate solution. A judgment is consistent if it selects the same winner across different prompt positions. Primacy/recency indicates a preference for the first/last candidate solution, respectively. Position inconsistent judgments suggest position bias and can be classified as primacy-preferred or recency-preferred.

To enhance the understanding of position bias and LLM-as-a-Judge, we propose a comprehensive framework designed to dissect and analyze position bias **within the context of pairwise comparative assessment**. We focus on this context because LLM judges demonstrate superior performance in making pairwise comparisons over other settings like pairwise scoring (Zheng et al., 2024b; Liusie et al., 2024), making it a practically preferable choice for investigations.

Our contribution can be summarized as following: (1) introducing a systematic framework for evaluating LLM-as-a-Judge's position bias in terms of **Repetition Stability**, **Position Consistency**, and **Preference Fairness**; (2) identifying Judge-level, Candidate-level, and Task-level factors that affect position bias; and (3) providing insights into future benchmark design and paving the way for effective mitigation strategies for position bias.

## 2 METHODS & DEFINTIONS

LLM-as-a-Judge can be classified into two categories: *score-based* and *relation-based* assessments. Our study focuses on relation-based assessment, specifically under the *pairwise comparison setting*, for several reasons. First, the *pairwise paradigm* is a popular and classic scenario that has influenced a wide range of fields (Qin et al., 2024). Second, as noted by Liu et al. (2024), LLM and human evaluations are more aligned in the context of pairwise comparison than any of the score-based assessments. Additionally, numerous studies have shown that pairwise comparative assessment outperforms other judging settings in terms of position consistency(Zheng et al., 2024b; Liusie et al., 2024). Finally, pairwise comparisons can be extended to more complicated frameworks of relation-based assessment, such as listwise comparisons, through some advanced ranking algorithms (Qin et al., 2024; Liu et al., 2024). Therefore, analyzing position bias in this context can provide a deeper understanding of its fundamental nature.

In the context of pairwise comparative assessments, LLM judges are prompted to choose the better candidate solution to the task question. To accommodate the possibility of a tie, various **option modes** are introduced. Two-Option mode restricts judges to choosing the better solution between two options, labeled {A} for the first and {B} for the second as shown in Fig. 1. Three-Option mode adds an additional choice, {C}, allowing judges to indicate a tie if neither solution is preferable. Four-Option mode further expands the choices, with {C} representing a *both good tie* and {D} a *both bad tie*. In our experiments, we employed Three-Option mode for MTBench (Zheng et al., 2024b) and Two-Option mode for DevBench (Li et al., 2024), aligning with their respective original frameworks. These option modes are explicitly specified in the system prompts to clearly guide the decision-making process of the LLM judges.

To set up our experiment, as shown in Fig. 1, the system prompt, option mode, task question, and solutions from two candidate models (*original prompt*) are presented to the LLM judges to select the winning solution. The experiment is conducted in a double-blind setting. The identities of the candidate models are hidden from the LLM judges, and the candidate models are unaware that their solutions will be compared to another model when answering the question. Then, the prompt with solutions in a swapped position (*swapped prompt*) is given to the same judge again, which results in a judgment pair. As shown in Fig. 2, if the LLM judge consistently favors the same solution regardless of the swapped position, it is considered position consistent with a fair preference. Conversely, if the LLM judge selects different winners, position bias is observed, with the preference direction being either primacy or recency.

To systematically investigate position bias in pairwise comparative judging scenarios, we evaluate LLM judges using three key metrics: **Repetition Stability** ($RS$), **Position Consistency** ($PC$), and **Preference Fairness** ($PF$). While $PC$ has been widely adopted in prior research, we introduce $RS$ to validate the reliability of LLM judges and $PF$ to provide a more comprehensive assessment. Further analysis examines the impact of judge-level, candidate-level, and task-level factors on position bias. Therefore, for each judge, candidate, and task, we calculate $RS$, $PC$, and $PF$ to thoroughly explore the variance in position bias and its impacting factors.

### 2.1 EVALUATION METRICS

We validate the capable LLMs with a high repetition stability and then evaluate their position bias in terms of position consistency and preference fairness. The metrics are introduced as follows.

**Repetition Stability** ($RS$) evaluates the reliability of LLM judges when presented with identical queries multiple times. It is essential to determine whether the judgments of LLMs, and consequently the observations of position bias, stem from a consistent evaluation pattern or merely random variations. We measure this by calculating the percentage of the most frequent selections across multiple trials for each query, aggregated from all queries within each dataset. This metric is formalized as

$$RC = \frac{1}{N} \sum_{j=1}^{N} \frac{1}{n_j} \max_{\forall k \in S} \left\{ |C_k^j| \right\}, \quad S = \{A, B, C, ...\} \tag{1}$$

where $S$ refers to the set of choice options depending on the option mode, $|C_k^j|$ denotes the counts of each choice option selected by the judge for the $j^{th}$ query, $n_j$ represents the total number of repeating trials for that query, and $N$ is the total number of queries. The value of $RS$ ranges from a small positive value depending on the option mode, indicating completely random decisions, to 1.0, indicating perfect stability.

**Position Consistency** ($PC$) quantifies how frequently a judge model prefers the same solution after the order of solutions is swapped. It is calculated as the ratio of consistent evaluation pairs to the total number of valid evaluations, where a pair is deemed consistent if the judge model prefers the same winning solution. Formally, it is calculated as

$$PC = \frac{1}{n} \sum_{j=1}^{n} \mathbf{1}_{\{(C_{\text{original}}^j, C_{\text{swapped}}^j) \in V\}}, V = \{(A, B), (B, A), (C, C), ...\} \tag{2}$$

where $(C_{\text{original}}^j, C_{\text{swapped}}^j)$ denotes the judgment pair for the $j^{th}$ query, $n$ represents the number of prompt pairs, and $V$ is the set of choice pairs that correspond to position consistency. An example of such pairs of choices can be found in Fig. 2. This formula provides a direct measure of a judge model's position bias and has been widely used in previous studies for its simplicity.

**Preference Fairness** ($PF$) is another crucial characteristic of position bias measuring the extent to which judge models favor certain solution positions. In the pairwise comparative setup, an LLM judge may exhibit a preference for either primacy or recency. These terms replace the more verbose "preference for the first/second position" used in previous studies, ensuring clarity and generalization for future research. The examples of such preferences are demonstrated in Fig. 2.

Position consistent judgments are fair in preference. The measurements become more complex for position inconsistent cases. Previous studies proposed two common ways to measure the preference fairness. A straightforward way is to count the primacy-preferred and recency-preferred judgment pairs, which we termed as primacy-count-number ($pcn$) and recency-count-number ($rcn$). The counts are then normalized by the total number of prompt pairs (Zheng et al., 2024b; Zhu et al., 2023b). However, **the sensitivity of this measurement highly depends on the size of dataset**, making comparisons across datasets unreliable, especially when the number of questions and instances varies for each task.

Alternatively, instead of normalizing over the complete dataset, studies like (Li et al., 2023c; Liusie et al., 2024) treat position inconsistent evaluation instances independently. They calculate the percentages of primacy-preferred and recency-preferred judgment pairs relative to the total number of position inconsistent pairs. We denote these as *inconsistent primacy rates (ipr)* and *inconsistent recency rates (irr)*, where $ipr + irr = 1$. However, this approach is problematic because it overlooks the fact that "position consistent judgments are also position fair," which can lead to **overly penalizing highly consistent LLM-judges**. For instance, if only one out of a hundred pairs is inconsistent, this method would classify it as 100% primacy-preferred or recency-preferred. However, this outcome should be distinguished from cases where all one hundred pairs are inconsistent and biased toward one position, which would also result in 100% preference unfairness.

To overcome these limitations, we introduce a more granular and scalable measurement that combines the strengths of both methods, to assess preference fairness. The $PF$ score is formally calculated by

$$PF = \frac{PF_{raw} - S_{min}^-}{S_{max}^+ - S_{min}^-} \times 2 - 1, PF_{raw} = (rcn \times irr) - (pcn \times ipr) \tag{3}$$

where $S_{min}^-$ and $S_{max}^+$ are the minimum and maximum achievable $PF_{raw}$ scores for each judge on each task, respectively. This min-max scaling technique ensures comparability across datasets by accounting for the range of achievable scores and centering the scale around zero. Compared to prior methods, the use of $rcn$, $pcn$ weighted by $irr$ and $ipr$, and scaling across judges and tasks implicitly accounts for the position consistent judgments. Another advantage of this approach is that it quantifies preference fairness with a single score, rather than the two separate scores used in previous studies. The $PF$ score is interpreted as follows:

$$PF = \begin{cases} 1, & \text{if } PC = 0 \text{ and entirely recency-preferred} \\ x \in (0, 1), & \text{Recency-preferred} \\ 0, & \text{Preference Fair} \\ x \in (-1, 0), & \text{Primacy-preferred} \\ -1, & \text{if } PC = 0 \text{ and entirely primacy-preferred} \end{cases}$$

To summarize, our proposed preference fairness score is superior to prior measurements due to its ability to provide a single, comprehensive score that accounts for all evaluation instances, while maintaining its sensitivity across datasets, even when the number of questions and instances varies for each task.

## 2.2 FACTORS AFFECTING POSITION BIAS

Table 1: Categorization of the factors that influence position bias, where 'Judge' refers to the LLM-judge and 'Candidate' refers to the candidate model whose solutions to the task-specific questions are evaluated by the judges. Factors that have a significant impact on position bias are highlighted with an **asterisk (*)** by bidirectional stepwise regression tests and marked red by empirical findings. 'Task Input' refers to the question itself, while 'Task Output' refers to the candidate model's solutions to the question, making it both a Candidate-level and Task-level factor. 'Prompt' encompasses the complete query presented to the judges, including Task Input, Task Output, and system prompts.

| Factor | Judge-level | Candidate-level | Task-level |
|---|---|---|---|
| Familial Property | ✓* | × | × |
| Answer Quality Gap | × | ✓* | × |
| Task Input Length | × | × | ✓ |
| Task Output Length | × | ✓* | ✓* |
| Prompt Length | × | ✓ | ✓ |

To investigate the factors influencing position bias in LLM judges, we categorized these factors into three groups: **Judge-level**, **Candidate-level,** and **Task-level** factors. Each group includes specific factors, that we hypothesize, may impact position bias, which we explore through a series of experiments. Table 1 lists the five factors analyzed in this study. By design, our framework allows for the integration and assessment of additional influencing factors.

Among the influencing factors we identified, we chose "familial property" for Judge-level factors instead of model sizes and training specifics, which are often proprietary and not publicly accessible for the models involved in our experiments. The model family grouping reflects similar model sizes and training specifics of the model members. The familial categories of the models used in our studies are (1) *GPT*, (2) *Claude*, and (3) *Gemini* allowing for straightforward grouping by company and version. However, we also observed that all GPT-4 and later models, along with Claude-3.5-Sonnet, Claude-3-Opus, and Gemini-1.5-pro, form a group of the "most capable" models sharing similar judging capabilities. More details and discussions about the familial property can be found in Appendix Sec. C, where we examined the LLM mutual agreements to study the similarities and differences across families.

**Answer quality gap**, based on our findings, is highly impactful yet under-explored in prior research. We define the quality of a candidate's answer/solution by how effectively it addresses the question. Consequently, the answer quality gap refers to the disparity in quality between the solutions from two candidate models to the same question and is considered the Candidate-level (i.e., solution provider) factor. Ideally, when a reliable LLM judge is presented with a question and corresponding solution pairs, it would prefer the higher-quality solution, where the corresponding candidate is

denoted as the winner selected by the judge. Following this assumption, we measure the answer quality gap by the win rates between two candidates on a set of tasks and questions. We regard a tie judgment as 'half-win' for both candidates when calculating its win rate over another, depending on option mode. If position bias occurs, the winner may be inconsistent when the order of candidate solutions is swapped in the query. Therefore, we categorize the LLM judgments into three groups: cases where the same winner is consistently chosen in both the original and swapped queries (termed "consistent wins"), cases where both responses are deemed equal (termed "consistent ties"), and cases where different winners are selected after the solutions are swapped. We denote these counts as the number of consistent wins ($C_w$), consistent ties ($C_t$), and inconsistent judgment pairs ($C_I$), respectively. Following Zheng et al. (2024b), we count inconsistent judgment pairs also as ties, or 'half-wins', to ensure calculations are feasible in all cases for the subsequent computations.

Inspired by Zheng et al. (2024b), we define the **overall win rate** ($owr$) of a model's solution over the other as: $owr = \frac{1}{n}[C_w + \frac{1}{2}(C_t + C_I)]$, where $n$ is the total number of judgment pairs. Then the **answer quality gap** ($\delta_q$) is calculated as $\delta_q = |owr - 0.5|$, where 0.5 is the overall win rate when we only have ties. In contrast to using only consistent win rate (calculated as $\frac{C_w}{n_c}$, where $n_c$ is the number of position consistent judgment pairs) to quantify $\delta_q$ (Zheng et al., 2024b; Li et al., 2023b; Raina et al., 2024), the adoption of overall win rate incorporates all data points and captures the "comparable quality" cases, where responses in similar quality might lead to position biased judgments, a scenario that the consistent win rate might overlook. More discussions can be found in Appendix. Sec. D.

## 3 EXPERIMENTS

### 3.1 EXPERIMENT SETTINGS

In this study, we evaluated twelve up-to-date commercial models from the GPT (OpenAI, 2023), Claude (Anthropic, 2024), and Gemini (Gemini Team, 2024) series using our framework. Additionally, three Llama models (Touvron et al., 2023), with parameters ranging from 7B to 13B, were selected as open-source exemplars due to their popularity and recognized capability in reasoning tasks. However, due to their limited performance on our tasks, as shown in Table 2, we focused our investigation solely on more powerful closed-source models. We included the results of open-source models for completeness.

We adopted the modified MTBench (Zheng et al., 2024b) and DevBench (Li et al., 2024) datasets for our study due to their demonstrated high human-LLM agreement and the reliability of state-of-the-art LLMs on the evaluation tasks. We fixed one of the candidates as **vicuna-13b-v1.3** for MTBench and **human** for DevBench to serve as baselines, ensuring decent quality of responses.

MTBench consists of 30 candidate models, 8 tasks, and 10 questions per task; for DevBench, we divide the *general* metric into more detailed ones and consider them as different tasks, resulting in 10 candidate models, 14 tasks, and 8 questions per task. We then paired solutions of these candidate models with that of the baseline candidate for evaluation by the LLM judges. The prompt templates we used are identical to those used in the benchmarks. More details about the models, tasks, and prompts can be found in Appendix. Sec. H.

To compute repetition stability, we sampled 3 questions per task and 4 candidate models, paired with baseline candidates, for each judge to evaluate across 3 repetitive trials. This resulted in 576 instances per judge for MTBench and 432 instances per judge for DevBench. The *temperature* hyperparameter was set to 1 for all judge models to generate nontrivial results. To compute position consistency and preference fairness, the number of instances increased to 4,800 and 2,240, covering the entire MTBench and DevBench datasets. In total, approximately 100,000 evaluation instances were analyzed in this study.

To identify significant factors contributing to position bias, we performed bidirectional stepwise regression on data from the two benchmarks. We used variables such as average lengths of input, output, and prompt; answer quality gap; LLM judge series; candidate identities; and task categories to predict $PC$ and $PF$, respectively. Each model prunes non-significant variables based on the Akaike Information Criterion (AIC) score. This process involves both forward selection and back-

ward elimination, with each "step" testing whether including or excluding a variable improves the model's AIC value. Further details about the process can be found in Appendix. Sec. F.

## 3.2 EMPIRICAL RESULTS AND INTERPRETATIONS

The evaluation results of 12 close-source and 3 open-source models in terms of repetition stability, position consistency, and preference fairness on MTBench and DevBench are listed in Table 2. For each judge, we calculate its average $RS$, $PC$, and $PF$ across all candidates and tasks. For $RS$ and $PC$, higher values are preferable. A high $RS$ value is particularly important as a prerequisite for meaningful computations of $PC$ and $PF$, ensuring the LLM judge's choice patterns are not merely random variations. Fig. 3 and Fig. 7 demonstrate that position bias varies by judges and tasks significantly. Fig. 4 further investigates the impact of answer quality gap and lengths of prompt components on position bias. These analyses were conducted by considering all judges together on MTBench. The detailed analyses for each judge and on DevBench can be found in Appendix.Sec.E, and Sec. G.

Through bidirectional stepwise regression, we found that, based on data from the two benchmarks, LLM judge series, candidate identities, and task categories significantly impact Position Consistency among all variables. Similarly, these factors also contribute significantly to Preference Fairness. Additionally, we found that average output length is a statistically significant predictor of $PF$. This finding is not surprising, as longer outputs are generally perceived as higher quality and more preferred. Quantitative results and more discussions can be found in Appendix. Sec. F.

Table 2: Evaluation results for Repetition Stability ($RS$), Position Consistency ($PC$), and Positional Fairness ($PF$). Top 5 performances are marked in **bold**. "Error" is due to failure of judgment generation (e.g, exceeding context window) or invalid extraction using regular expression for the specified judgment output format. High error rates and low $RS$ are marked red, implying further evaluations (i.e., $PC$, and $PF$) are invalid and meaningless because of insufficient accessible data. Therefore, **open-source models (e.g., Llama) are not investigated in the paper** because of their high error rates under the difficulty and length of our evaluation instances. In contrast, up-to-date commercial models generally demonstrated minimal error and high repetition stability, making them more practical and meaningful for investigations. Chatbot Arean Leaderboard's overall ranking of the models is listed.

| Judge | MTBench | | | | DevBench | | | | Arena |
|---|---|---|---|---|---|---|---|---|---|
| | $RS$ | $PC$ | $PF$ | Error | $RS$ | $PC$ | $PF$ | Error | |
| Caude-3.5-Sonnet | $0.96 \pm 0.07$ | $\mathbf{0.82 \pm 0.14}$ | **0.01** | 0.00 | $0.95 \pm 0.09$ | $0.76 \pm 0.16$ | 0.22 | 0.00 | 7 |
| Claude-3-Opus | $0.95 \pm 0.08$ | $0.70 \pm 0.19$ | 0.22 | 0.00 | $0.96 \pm 0.07$ | $0.69 \pm 0.20$ | 0.29 | 0.00 | 17 |
| Claude-3-Sonnet | $0.93 \pm 0.11$ | $0.59 \pm 0.22$ | 0.32 | 0.01 | $0.95 \pm 0.09$ | $0.71 \pm 0.22$ | 0.23 | 0.00 | 37 |
| Claude-3-Haiku | $0.89 \pm 0.18$ | $0.57 \pm 0.18$ | 0.18 | 0.00 | $0.90 \pm 0.17$ | $0.23 \pm 0.14$ | 0.75 | 0.00 | 46 |
| Gemini-1.5-pro | $0.97 \pm 0.09$ | $0.62 \pm 0.19$ | 0.23 | 0.03 | $0.87 \pm 0.17$ | $\mathbf{0.84 \pm 0.17}$ | **0.03** | 0.13 | 12 |
| Gemini-1.5-flash | $1.00 \pm 0.00$ | $0.67 \pm 0.17$ | 0.07 | 0.00 | $0.04 \pm 0.08$ | $0.92 \pm 0.39$ | 0.00 | 0.96 | 25 |
| Gemini-1.0-pro | $0.89 \pm 0.18$ | $0.57 \pm 0.18$ | 0.30 | 0.00 | $0.85 \pm 0.26$ | $0.66 \pm 0.20$ | **-0.05** | 0.00 | 66 |
| o1-mini | $0.90 \pm 0.07$ | $\mathbf{0.76 \pm 0.15}$ | **-0.04** | 0.00 | $0.93 \pm 0.12$ | $\mathbf{0.84 \pm 0.13}$ | **-0.07** | 0.00 | 2 |
| GPT-4o | $1.00 \pm 0.02$ | $\mathbf{0.76 \pm 0.18}$ | -0.12 | 0.00 | $0.98 \pm 0.03$ | $\mathbf{0.80 \pm 0.16}$ | **-0.12** | 0.00 | 6 |
| GPT-4-Turbo | $0.94 \pm 0.10$ | $\mathbf{0.75 \pm 0.16}$ | **0.02** | 0.00 | $0.97 \pm 0.06$ | $\mathbf{0.79 \pm 0.18}$ | 0.16 | 0.00 | 17 |
| GPT-4 | $0.97 \pm 0.05$ | $\mathbf{0.82 \pm 0.15}$ | **0.02** | 0.00 | $0.97 \pm 0.05$ | $\mathbf{0.83 \pm 0.15}$ | -0.13 | 0.00 | 52 |
| GPT-3.5-Turbo | $0.96 \pm 0.07$ | $0.70 \pm 0.18$ | **0.06** | 0.00 | $0.99 \pm 0.02$ | $0.76 \pm 0.18$ | **-0.02** | 0.00 | 94 |
| Llama-2-13B | $0.45 \pm 0.22$ | $0.56 \pm 0.33$ | -0.08 | 0.55 | $0.00 \pm 0.00$ | $0.00 \pm 0.00$ | 0.00 | 1.00 | 99 |
| Llama-2-7B | $0.83 \pm 0.15$ | $0.44 \pm 0.19$ | 0.11 | 0.17 | $0.00 \pm 0.00$ | $0.00 \pm 0.00$ | 0.00 | 1.00 | 109 |
| Llama-3-8B | $0.16 \pm 0.18$ | $0.43 \pm 0.39$ | -0.05 | 0.84 | $0.06 \pm 0.09$ | $0.04 \pm 0.09$ | -0.05 | 0.94 | 57 |

## 4 MAIN FINDINGS

**Position Bias of Capable Judges are not Mere Random Variations** As shown in Table 4, the capable judges on the benchmark tasks, supported by minimal "Error" rates, generally exhibit $RS$ values above 0.85. The most up-to-date and capable models, such as Claude-3.5-Sonnet, Claude-3-Opus, GPT-4, and GPT-4o, all achieve near-perfect $RS$ scores exceeding 0.95 on both benchmarks. These results confirm that LLM judgments, and the resulting position bias, are not merely random

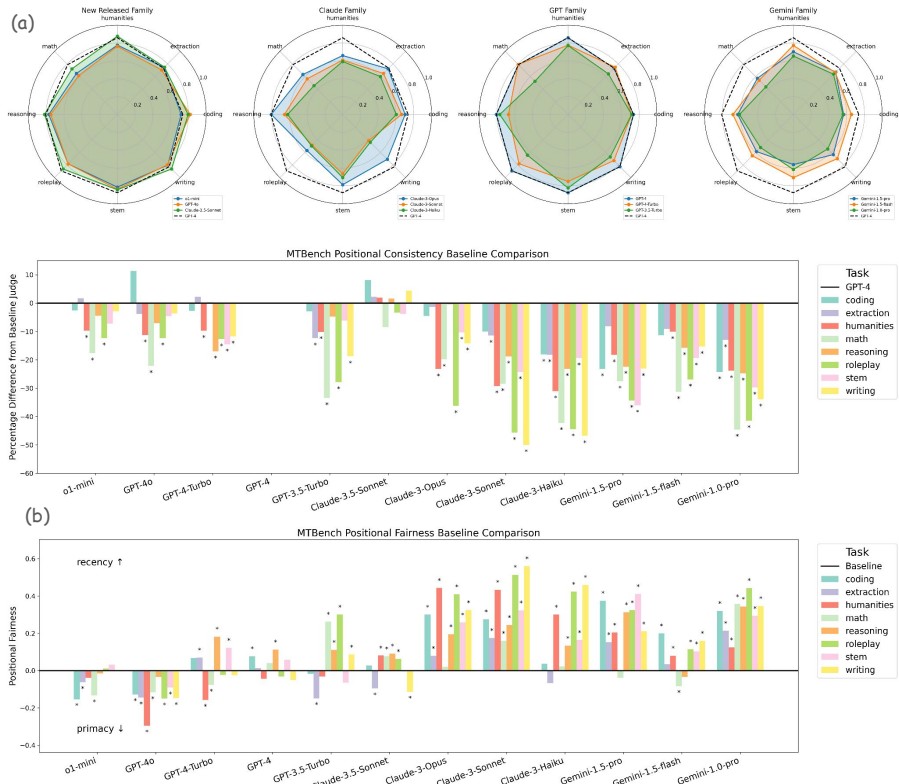

Figure 3: Baseline comparisons of judges across tasks on MTBench. An asterisk marks the statistical significance by Student's t-tests. Figure (a) includes a radar chart comparison by family and an overall baseline bar chart comparison of $PC$. The baseline model is chosen to be GPT-4 for its generally superior performances. Figure (b) demonstrates the $PF$ score of evaluated judges across tasks where the baseline is $PF = 0$.

variations. This strengthens confidence that one-time generated judgments by these validated LLMs accurately reflect their judging capabilities.

**Position Bias Varies by Judge & Task** As shown in Table 2, in general, GPT-4 excels as the best-performing judge model across metrics and benchmarks, hence considered the baseline model in Fig. 3 for $PC$ comparisons. However, certain models achieve comparable or superior performances than GPT-4 on certain tasks. For example, according to Fig. 3, for *coding* task evaluation, GPT-4o and Claude-3.5-Sonnet are likely more ideal judges. Also, GPT-3.5-Turbo achieves comparable $PC$ as GPT-4, indicating that it may be employed as a cost-effective alternative to coding evaluations.

Additionally, significant variations of $PF$ across tasks are observed. The judges achieving close-to-0 $PF$ in general, such as GPT-4 and Claude-3.5-Sonnet, exhibit varied preference directions across tasks, preferring primacy on some tasks while recency on the others. Particularly, o1-mini, while being primacy-preferred on coding, extraction, and math, exhibits almost fair preferences on reasoning, role play, and writing tasks. Even for uniformly recency-preferred judges such as Claude-3's and Gemini-pro's, the extent of biased preference, as reflected by $PF$ values, varies by task.

Moreover, high position consistency does not guarantee fairness. For example, on coding task evaluations, GPT-4 and GPT-4o achieve the top consistency but are significantly recency-preferred and primacy-preferred, respectively. In comparison, GPT-3.5-Turbo is highly preference fair while having comparable consistency.

Therefore, the position bias of LLM judges is judge-dependent and task-dependent. This observation is confirmed by the bidirectional stepwise regression where judge identities and task categories are statistically significant predictors of $PC$ and $PF$. In practice, to achieve the most reliable evaluation results, one needs to select task-specific LLMs whose consistency and fairness are in

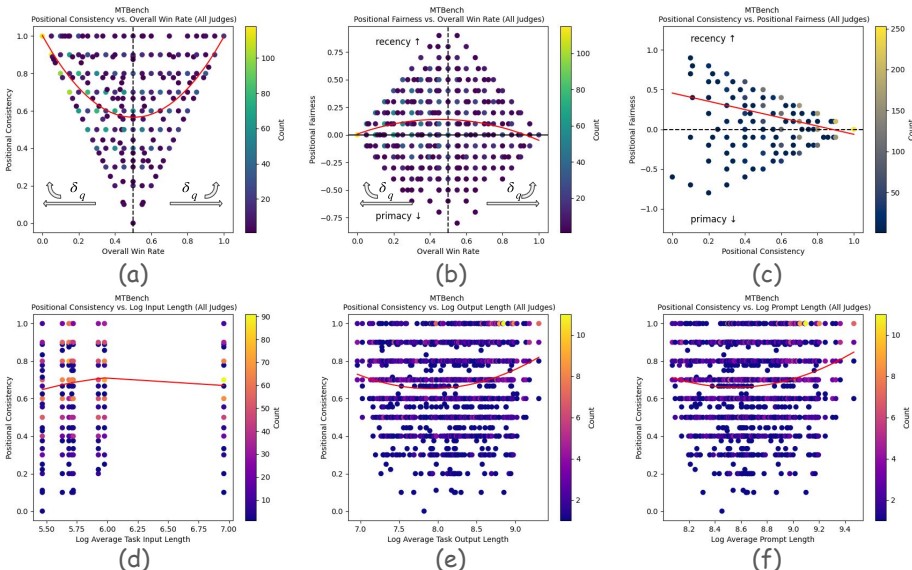

Figure 4: Impact of answer quality gap and lengths on position bias. All figures are based on the integration results of all judge models on MTBench. Figures (a) and (b) investigate $PC$ and $PF$ vs. overall win rate, hence the answer quality gap, respectively. Generally, judgments become more consistent and fair as the quality gap decreases. Figure (c) demonstrates the relationship between $PC$ and $PF$, where higher consistency is likely to indicate better fairness. Figure (d)(e)(f) plots the relationship between lengths of prompt components and $PC$. The irregular and inconclusive patterns imply that position bias is not significantly impacted by the length of prompts and their components.

balance. Moreover, as shown in Table 2, a high ranking in the overall reasoning performance of a model does not guarantee its judging reliability. Similar findings can be observed in the DevBench results, as detailed in Appendix.Sec. G.

**Position Bias Correlates to Answer Quality Gap** Intuitively, the difficulty of judging a pair of candidate answers is largely reflected by their difference in quality. In this study, as defined in Section 2.2, we quantify the quality gap ($\delta_q$) between answer pairs by the overall win rate ($owr$). Since an absolute tie has 0.5 $owr$, $\delta_p$ increases as $owr$ extends from 0.5 to 0 or 1. Fig. 4(a) exhibits a significant parabolic shape, indicating that $PC$ is positively proportional to $\delta_q$. This aligns with our intuition that the answer pairs with larger quality disparities are easier to achieve consistency, whereas those of similar quality are difficult to judge, increasing the likelihood of position bias that leads to lower $PC$. The same relationship is observed for each individual judge and across benchmarks, as demonstrated in Appendix.Sec. E.

Similarly, as shown in Fig. 1(b), judgments generally become more preference fair as $\delta_q$ increases. However, the extent is not as significant as for $PC$. Also, the relationship varies by judge, as some LLMs maintain preference fairness regardless of $\delta_q$. For example, as shown in Fig. 6, $PF$ of GPT models centered closely around 0 consistently, whereas that of Claude and Gemini-pro models exhibit a conspicuous proportional relationship on MTBench.

These observations align with the right-arrow shape as demonstrated in Fig. 4(c), where there is a general trend that judgments become preference fairer as position consistency increases. It also justifies the reasonableness of our quantification of preference fairness, as highly position consistent judges are not overly penalized and a perfect $PC$ should result in $PF = 0$.

Together, we conclude that as the answer quality gap enlarges, judges **generally** become more position consistent and preference fair according to the regression curves. However, exceptions are common, as shown by the individual scatter points of these figures. This indicates that though the answer quality gap significantly influences the position bias of LLMs, other factors also play important roles. Therefore, built on our findings, future studies may have better control over the answer

quality gap when evaluating LLM judges, exploring other impacting factors on position bias, and seeking potential mitigation strategies.

Our investigation also helps to explain the observed verbosity/length bias (Liu et al., 2024; Kim et al., 2023; Zheng et al., 2024b) and self-enhancement bias (Li et al., 2023c; Zheng et al., 2024b) of LLM judges. From our analysis, LLMs may prefer longer answers (verbosity/length bias) since more capable models usually generate longer solutions with higher qualities. Similarly, in the cases where the same LLM serves as both a candidate and a judge model, judges preferring the solutions generated by themselves (self-enhancement bias) are usually the ones who generate responses of higher quality (e.g., GPT-4). In both cases, the observed biased preference may be fundamentally due to a prominent answer quality gap instead of length or self-identity.

**Position Bias is weakly Length-dependent** We investigate the impact of three different lengths on the position bias of LLM judges: the length of the question (task input length), the solution length of candidate models (task output length), and the length of the entire prompt (prompt length). Since the tokenization methods vary by models, we apply the length of the prompt component strings for uniformity and scalability. The analysis from Fig. 4(d)(e)(f) shows that there is very weak relationship between the lengths of prompt components and position bias. By stepwise regression, the average task output length is only significant in predicting $PF$, adding a minimal change in AIC as shown in Appendix Table. 4. Therefore, the impact of lengths on position bias is minimal compared to answer quality gap and judge or task level variances.

**Practical Implications for Benchmark Design** We complement our investigations of position bias with a LLM agreement/disagreement analysis in Appendix.Sec. C to gain further insights. In particular, the disagreement analysis helps identify instances where it is easy or difficult for judges to reach a consensus, reflecting the complexity of the judgment task. Based on our findings regarding the answer quality gap and LLM agreement/disagreement, this study offers practical implications for designing evaluator benchmarks that control the difficulty of judging tasks. Our findings show that the hardest-to-judge instances are those where (1) LLMs frequently disagree with each other, (2) comparable win rates and minimal quality gaps exist between the two candidate models, and (3) significant position bias is exhibited by the majority of judges.

One qualitative analysis example of a difficult-to-judge instance comes from MTBench question 120, which involved a math problem where both candidate models, Vicuna-13b-v1.3 and WizardLM-30b, provided incorrect solutions. The problem was: (1) Given $f(x) = 4x^3 - 9x - 14$, find the value of $f(2)$, and (2) Find $x$ such that $f(x) = 0$. Vicuna-13b-v1.3 incorrectly applied the quadratic formula to solve the cubic equation, while WizardLM-30b used an enumeration method but failed to identify the correct answer. This resulted in a 4{A}-3{B}-5{C} choice pattern among the twelve judges, illustrating a typical case of significant disagreement.

We observed that some judges exhibited hallucinations, making false positive judgments or ignoring the reference answer. Others preferred either correctness (choosing {C} as a tie) or methodological soundness (choosing {B}, favoring WizardLM-30b). In the swapped order setting, 9 out of 12 judges displayed position bias. This highlights the challenge of judging solutions with minimal quality gaps, leading to greater position bias and subjectivity.

The key takeaway is that reducing hard-to-evaluate, highly subjective instances can improve benchmark design. Likewise, minimizing trivial cases where judges easily agree would enhance the benchmark's robustness. Clearer and more specific evaluation criteria could also help resolve ambiguity in difficult instances.

## 5 CONCLUSION

In conclusion, this paper proposes a systematic, scalable framework for evaluating the position bias of LLM judges in pairwise comparative scenarios in terms of repetition stability, position consistency, and preference fairness. Through comprehensive evaluations of 12 judges on two benchmarks across 22 tasks with over 100,000 evaluation instances, we observe significant variations of position bias across judges and tasks. We also discover that position bias is weakly length-dependent but significantly impacted by the answer quality gap. Our findings enhance the understanding of position bias, paving the way for more effective mitigation strategies and more robust and reliable evaluation systems.

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

APPENDIX TABLE OF CONTENTS

APPENDIX

## A  CONTRIBUTION & LIMITATION

**Contributions**  Despite the limited number of models, tasks, and judging types, our proposed systematic framework is agnostic to these criteria and can be used with any LLM judge. Our work is particularly valuable for validating judges in terms of repetition stability and then evaluating them through position consistency and preference fairness. We also explore the factors influencing position bias and their quantitative impacts, such as the answer quality gap and lengths of prompt components. Particularly, our study is the first to establish a direct connection between quantified answer quality disparities and position bias, highlighting the urgent need for a comprehensive understanding of how position bias affects LLM judges. Additionally, the mutual agreement and disagreement analysis of LLM judges are insightful for identifying the difficulty of evaluation instances and beneficial for benchmark assessment and designs. All findings from our study enhance the understanding of the position bias of LLM-as-a-Judge, paving the way for more effective mitigation strategies and more robust and reliable evaluation systems.

**Social Benefits**  Our comprehensive understanding of position bias in LLM-as-a-Judge provides multiple benefits to the community. First, it enhances the reliability and trustworthiness of LLM judges in terms of repetition stability validation. Our focus and investigation on preference fairness inspire future works to improve the fairness of LLM evaluators simultaneously with consistency and reliability. Moreover, our emphasized trade-offs between position consistency, fairness, and practical applicability when selecting task-specific LLM judges provide crucial judge-selection guidance for future work. Additionally, the baseline comparison analysis across judges and tasks offers reasonable judge model recommendations for more reliable, consistent, and fair evaluation results. Besides, the LLM agreement/disagreement analysis in Appendix.Sec.C 5 provides an applicable tool to identify the difficulty of judging certain instances. Together with findings from the significant impact of the answer quality gap on position bias, our study benefits future benchmark designs and evaluations and lays the groundwork for potential mitigation strategies of position bias. Overall, our study contributes to a more reliable, unbiased evaluation system that benefits the application of LLM evaluators across diverse fields, such as healthcare (Chen et al., 2024c), instruction following and prompting (Zeng et al., 2023; Chua et al., 2024), multimodal assessment (Chen et al., 2024a), and recommender systems (Hou et al., 2024).

**Strength**  Our framework of systematically understanding position bias has the following strengths: (1) scalability and ease of use - our proposed framework is scalable to various judges, models, tasks, and judging types. It is also straightforward and convenient to implement, facilitating a more comprehensive evaluation of LLM judges for future works. (2) comprehensive experiments - we investigate the position bias of 12 LLM judges on 2 benchmarks across 22 tasks and around 40 answer-generating models in terms of repetition stability, position consistency, and preference fairness, resulting in more than 100,000 evaluation instances; the exploration of the factors impacting position bias from Judge-level, Model-level, and Task-level and the findings from LLM agreement analysis provide additional insights to enhance the understanding of position bias. (3) accurate measurement - we formally define crucial metrics ($RS$ and $PF$) for evaluating LLM judges' position bias that are underexplored by prior works and offer detailed explanations for their rationality and rigorousness.

**Limitations**  Due to computational limits, our study can be potentially extended to provide more comprehensive insights. (1) We only study the pairwise comparative assessment of LLM-as-a-Judge for its best consistency, ease of use, and scalability, but more types of judging can be explored, such as pairwise scoring and listwise ranking. (2) We only study 12 up-to-date commercial LLM judges for systematic investigations. Llama 7b-13b models that we choose as representatives of open-source models of similar sizes, face extreme difficulty in judging the instances explored in this study. Due to limited resources and considering practicality, we did not experiment with open-source models with larger sizes. However, further exploration may include analysis of these state-of-the-art open-source and fine-tuned judge models using our proposed framework. (3) We only study position bias on two benchmarks, MTBench and DevBench, which is still limited despite 22 tasks and 40 answer-generating models. Also, the number of questions for each task is limited. Further studies can be extended to more benchmarks to have a broader understanding of position bias across benchmarks,

tasks, and questions. (4) Due to data accessibility, we are unable to investigate Judge-level factors in terms of model parameter sizes and architectures, which we expect will have a significant impact on position bias. Alternatively, we consider the family property to account for Judge-level factors by grouping models with similar properties (e.g., approximate parameter size) together. We also investigate variances of position bias across individual judges. However, future exploration on these model properties, especially those available on open-source models, will provide more insights.

**Future Work** Besides addressing the limitations, the following future work, built on findings from our study, could provide more insights: (1)prompt setting - in this study, we apply the exact default prompt settings of MTBench and DevBench. However, the positional order and prompt style of not just the model-generated responses but also system prompt components (e.g., agent role assignment, mode of judging, direct mention of not making biases) may also have an impact on the extent to which LLM judges make position bias. (2) open-source insights: Further investigations of open-source and fine-tuned models on easier-to-evaluate benchmark may provide more insights into the training stages or model structures that are affecting or causing the position bias. Then our framework offers better control on the other impactful factors such as the answer quality gap to make these further investigations more accurate.

# B RELATED WORK

## B.1 LLM-AS-A-JUDGE

In recent years, Large Language Models (LLMs) have emerged as a transformative technology, garnering global attention and stimulating substantial research into their applications. For evaluative tasks, particularly subjective ones, human assessment is considered the gold standard due to its comprehensive and open-ended nature (Zeng et al., 2023). However, it lacks scalability and reproducibility (Karpinska et al., 2021).

As a result, LLMs have increasingly been used as substitutes for human evaluators across various Natural Language Generation (NLG) domains and tasks (Zheng et al., 2024b; Li et al., 2024; Chiang & Lee, 2023b; Liusie et al., 2024; Chen et al., 2023; Fu et al., 2023; Wang et al., 2023a; Kocmi & Federmann, 2023b; Ji et al., 2023; Liu et al., 2024; Zeng et al., 2023; Liu et al., 2023; Shen et al., 2023; Kocmi & Federmann, 2023a; Chiang & Lee, 2023a; Karpinska et al., 2021; Dubois et al., 2024; Kim et al., 2023; Liang et al., 2023; Chang et al., 2023; Zhong et al., 2023; Chia et al., 2023; Deb & Das, 2023; Ye et al., 2024; Min et al., 2023; Li et al., 2023d;c;b; Lu et al., 2024), including open-ended story generation (Chiang & Lee, 2023a), adversarial attacks (Chiang & Lee, 2023b), summarization (Karpinska et al., 2021), machine translation (Kocmi & Federmann, 2023a), and instruction following (Zeng et al., 2023).

These LLM evaluators, known as LLM-as-a-Judge, have attracted significant interest within both academic and industrial circles (Zheng et al., 2024b;a; Pezeshkpour & Hruschka, 2023; Chiang & Lee, 2023b; Shen et al., 2023; Liusie et al., 2024; Qin et al., 2024; Chen et al., 2023; Huang et al., 2024; Zhu et al., 2023b; Raina et al., 2024; Chen et al., 2024b; Wang et al., 2023b; Liu et al., 2024; Zeng et al., 2023; Khan et al., 2024; Kim et al., 2023; Li et al., 2023b;d; Shi et al., 2024; Li et al., 2023c; Chen et al., 2024c; Chua et al., 2024; Chen et al., 2024a). As LLMs have made content generation significantly easier, the volume of generated responses has increased, making it impractical to rely solely on human evaluation. Therefore, cost-effective LLM judges are needed to assess these responses efficiently. LLM-as-a-Judge is typically employed in Q&A evaluation tasks, where the LLM judge is prompted to evaluate the quality of responses, usually generated by models, answering the questions.

In many of these tasks, LLM judges have shown a high level of agreement with human evaluators (Zheng et al., 2024b; Li et al., 2024; Zhu et al., 2023b), yet in some tasks, they are less effective, largely due to inherent biases (Shen et al., 2023; Chiang & Lee, 2023b; Zheng et al., 2024a). Even in cases where agreement is high, judgments may still suffer from biases.

When employing LLM-as-a-Judge, various types of judging are available, which can be categorized either by the scale or comparative method. From the scale perspective, LLM-as-a-Judge can involve pointwise, listwise, or pairwise assessment (Qin et al., 2024). By comparative method, it can be either score-based or relation-based (Li et al., 2023d).

For example, pointwise scoring (Kim et al., 2023) lets the LLM judge score the response of one model to a question at a time based on some evaluating metrics. Pairwise/Listwise Scoring (Karpinska et al., 2021; Zhu et al., 2023b; Chen et al., 2024a; Zheng et al., 2024b) prompts the LLM judges to score a pair/list of model-generated answers. Listwise Ranking is another relation-based assessment that, instead of giving a score, requires the LLM judge to rank a list of responses following some specified order (e.g., from best to worst). Pairwise comparative assessment (Karpinska et al., 2021; Chen et al., 2024a; Zheng et al., 2024b), on the other hand, asks the LLM judge to select the superior response between a given pair, usually conducted in a double-blind manner: the generating model remains unknown to the judge, and the judging model remains anonymous to the answer generator.

Except for pointwise evaluation, all forms of LLM-as-a-Judge suffer—or are susceptible to suffering—from position bias due to the intrinsic nature of *position* and *comparison* within the prompt structure.

## B.2 Position Bias

Previous studies have discovered and investigated multiple types of biases, such as position bias (Zheng et al., 2024b; Li et al., 2024; Chen et al., 2024a; 2023; Chia et al., 2023; Chiang & Lee, 2023b; Khan et al., 2024; Zhu et al., 2023b; Zhong et al., 2023; Zheng et al., 2024a; Deb & Das, 2023; Zeng et al., 2023; Wang et al., 2023c; Qin et al., 2024; Wang et al., 2023b; Shi et al., 2024; Liusie et al., 2024; Liu et al., 2024; Kim et al., 2023; Karpinska et al., 2021; Hou et al., 2024; Kocmi & Federmann, 2023b; Pezeshkpour & Hruschka, 2023; Shen et al., 2023; Chiang & Lee, 2023a; Kocmi & Federmann, 2023a; Chen et al., 2024b; Li et al., 2023c;d; Raina et al., 2024), verbosity/length bias (Zheng et al., 2024b; Liu et al., 2024; Kim et al., 2023), self-enhancement bias (Li et al., 2023c; Zheng et al., 2024b), selection bias (Zheng et al., 2024a; Pezeshkpour & Hruschka, 2023), and contextual bias (Liu et al., 2024; Zhou et al., 2024).

Among these, **position bias** stands out as particularly significant, permeating a wide array of tasks and affecting judge models, including open-source (Li et al., 2023b; Chen et al., 2024c), proprietary commercial ones (OpenAI, 2023; Anthropic, 2024; Gemini Team, 2024), and fine-tuned models (Huang et al., 2024; Wang et al., 2023c; Zhu et al., 2023b; Li et al., 2023b; Kim et al., 2023).

To clarify, the aforementioned and following position bias refers to the concept within the context of LLM-as-a-Judge, meaning that LLM judge tends to favor responses based on their position in the prompt rather than their content. For example, in a **pairwise comparative assessment** scenario, if the LLM judge consistently selects the first response as superior even after switching the order of the two responses (same position, but different content), then a position bias occurs. Some other studies also use the term position bias (Ko et al., 2020; Aslanyan & Porwal, 2019; Blunch, 1984; Raghubir & Valenzuela, 2006), but in this research, our interest lies solely in the position bias specific to LLM-as-a-Judge.

Position bias is arguably the most prevalent and impactful bias among all. Chua et al. (2024) notes that their Bias-Augmented Consistency Training (BCT), an unsupervised fine-tuning scheme designed to promote consistent reasoning across prompts with and without biasing features, improves Chain-of-Thought (Wei et al., 2022) performance over self-training controls for all biases except position bias. Furthermore, Khan et al. Khan et al. (2024) point out that LLM judges are less confident when exhibiting position bias, and addressing this bias is highly complex due to varying confidence levels across judges and tasks.

Moreover, there is ongoing debate over whether selection bias originates from position bias. Pezeshkpour & Hruschka (2023) argue that LLMs are sensitive to the ordering of options in Multiple Choice Questions (MCQs), confirming that position bias contributes to this sensitivity. Raina et al. (2024) contest this view, asserting that selection bias stems less from position bias and more from token bias, which represents an inherent challenge for LLMs and contributes to poor robustness.

All of these findings underscore the critical importance of addressing the issue of position bias when employing LLM-as-a-Judge.

### B.3 DEAL WITH POSITION BIAS

Intuitively, position bias emerges because LLMs are sensitive to changes, especially positional changes, in prompts (Zheng et al., 2024b; Raina et al., 2024; Pezeshkpour & Hruschka, 2023; Zhao et al., 2021; Zhu et al., 2023a). Also, LLM judges are vulnerable to attacks (Raina et al., 2024; Chen et al., 2024b; Shi et al., 2024).

**Parse data with position bias** There are many ways to deal with the position bias. The naive way is to exclude the inconsistent judgments(i.e. if the LLM judge gives a positionally biased judgment on a pair or list of model responses) (Zheng et al., 2024b; Chen et al., 2024b; Wang et al., 2023b; Li et al., 2023b). While this ensures consistent and reliable remaining judgments, it does not resolve the fundamental problem. Moreover, if the LLM judge is highly biased, this method discards valuable evaluation instances and information, making it an ineffective and somewhat desperate measure.

To take the positionally inconsistent evaluations into account, one may either take an average for scoring-based judging (Zheng et al., 2024b; Raina et al., 2024; Li et al., 2023b; Wang et al., 2023b) or regard inconsistency as a *half-win* or *tie* for relation-based judging (Zheng et al., 2024b; Li et al., 2023c) after swapping the order of model responses in the prompt. For instance, in a pairwise scoring scenario, if model A receives a score of 8 when put in the first position and 4 when put in the second position, the overall score for it compared to model B would be 6; in the pairwise comparative case, if model A wins when put in the first position but then lose when put in the second position, it counts as a tie or half-win for both model A and model B. The latter way of *swapping + tie* is proposed because intuitively the position bias is more likely to occur when the model responses get evaluated share a similar quality in terms of the evaluating metric. Our quantitative study also verifies this intuition, evidenced by the fact the LLM judge's positional consistency (the percentage of positionally consistent pairs of judgment) is positively proportional to the **answer quality gap** between the model responses.

To save the expense of running the experiments more than once using swapped order, in practice, many of these studies also support a *random-shuffle* option in their code settings such that the baseline model for comparison does not remain in a fixed position.

**Solution attempts** Due to the significance of position bias, more sophisticated and advanced approaches have emerged to solve position bias as well, including bootstrapping (Hou et al., 2024), split-and-merge (Li et al., 2023d), and multi-agent discussion (Li et al., 2023c; Khan et al., 2024). However, these methods are either costly and time-consuming (e.g., multi-agent discussion and review to reach agreement) or ineffective.

Furthermore,Liu et al. (2024) suggests that existing calibration techniques designed to reduce bias, including context decomposition (Li et al., 2023d), order permutation (Wang et al., 2023b; Zheng et al., 2024a), ensembling (Li et al., 2023a), and batch calibration (Zhou et al., 2024), are insufficient to align LLM evaluators, even with supervised data.

Thus, position bias is a pervasive, significantly impactful, and challenging problem to solve.

**Understand before solving** We observe that the existing methods maybe ineffective or not enough satisfactory because there is a lack of understanding of the position bias. Although a variety of studies have researched this type of bias in LLM-as-a-Judge, a comprehensive understanding of what factors affect the position bias remains underexplored. In other words, without clarity on the key factors and their quantitative impact on position bias, the efficacy of current and future methods remains uncertain.

For instance, Li et al. (2023d) proposed a PORTIA approach that addresses the position bias to a large extent, receiving a 80.99% Fixed Coverage (the percentage of positionally inconsistent original assessments that are later corrected by PORTIA) for GPT-4 in a relation-based evaluation on 8 MTBench answer pairs, improving consistency from 93.44 % to 97.03%. However, the extraordinarily high consistency of the original evaluation may imply that the choice of answer pairs may be biased in terms of **answer quality gap**, a factor that our study proves to be significantly impactful. In other words, if the quality between the two model responses differs considerably, the LLM judges will make little position bias during judgment and hence is easy to calibrate. This example illustrates how lacking a comprehensive understanding of the factors affecting position bias can lead

to overestimating or misjudging the effectiveness of methods proposed to resolve it and improve the performance of LLM-as-a-Judge.

**Missing analysis for position bias** Besides, the positional preference direction aspect of position bias remains underexplored. Positional preference refers to the specific positions an LLM judge favors when position bias is evident. For instance, if an LLM judge consistently favors the responses that appear first in the prompt, it exhibits a preference for the first position. Our study denotes primacy-preferred and recency-preferred to describe biases toward the first and second positions, respectively. We integrate the measurement by formally defining *positional fairness* score to quantify the extent of a judge model's preference towards certain positions.

While prior research includes some positional preference results (Zheng et al., 2024b; Wang et al., 2023b; Zhu et al., 2023b; Li et al., 2023c; Liusie et al., 2024), none have conducted a detailed analysis. They focus primarily on positional consistency in relation to position bias, while we regard positional preference as an equally crucial component that needs to be examined, understood, and enhanced in future work.

Therefore, we propose that mitigating the position bias of LLM-as-a-Judge requires a simultaneous improvement in both positional consistency and positional fairness. A positionally consistent (high positional consistency) and positionally fair (nearly equal preference on primacy and recency when position bias occur) LLM judge is undoubtedly preferable. However, a trade-off between consistency and fairness/preference often occurs in practice. Previous studies that have provided results on positional preference have all shown that even state-of-the-art (SOTA) LLM judges struggle with both positional inconsistency and clear positional preferences.

Apart from that, previous studies have yet to examine how stable LLM judges are across repetitions. MLLM-as-a-Judge (Chen et al., 2024a) conducts repeated experiments but then takes an average/mode of the judgment only to make the evaluations more robust and reliable, without a focus on the repetitively stable judgments. In comparison, we investigate the repetition stability to determine whether position bias is solely influenced by the prompt's positional information and the judge's intrinsic properties or can be partly attributed to judgment randomness. If the judgments exhibit randomness over repetitions, then the position bias overlaps with repetition bias, complicating the issue and potentially invalidating previous findings.

### B.4 SUMMARY OF PRIOR WORK

To summarize, LLM-as-a-Judge has a large potential in alternating human judges on a wide range of tasks, in particular subjective evaluation, owing to its cost-effectiveness, high agreement with human judgments, reproducibility, and scalability. However, it suffers from various biases, most notably position bias, which is prevalent across different evaluation tasks, models, and judgment types. Not only is it a type of bias that significantly hinders the improvement and promotion of applying LLM-as-a-Judge, but it is also difficult to solve due to its complexity and lack of understanding in the community.

Existing solutions proposed to address this issue may be ineffective or uncertain in efficacy due to the unclear impact and nature of influencing factors. Additionally, whether the position bias is an essential bias or mere random variation has not been validated from prior works. The positional preference side, reflecting the positional fairness of the LLM judge, also requires comprehensive analysis. In a nutshell, a comprehensive understanding of the position bias of LLM-as-a-Judge is crucial to validate existing and future approaches to address this important problem.

### C LLM AGREEMENT ANALYSIS

Besides the exploration of position bias with a broad lens by average $PC$ and $PF$, instance-wise agreement between LLM judges is also insightful. Even two judges with the same $PC$ and $PF$ scores may not reach consensus on each instance. Therefore, this session investigates (1) what percentage of a set of evaluations do two LLM judges agree on each other? (2) how do the choices of all judges on an instance vary?

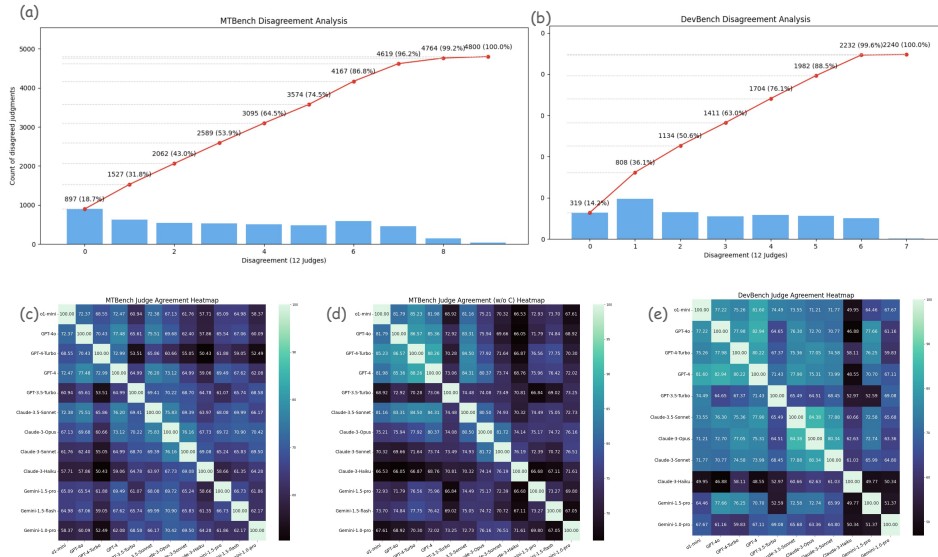

Figure 5: Disagreement Analysis and Mutual Agreement Heatmap of LLM judges on MTBench and DevBench. The blue bars of Figures (a) and (b) represent the number of instances with certain disagreement values, and the red cumulative curve displays the cumulative count and corresponding percentage over the benchmark dataset. The heat maps of Figures (c)(d)(e) marked higher mutual agreement values with a brighter color.

## C.1 MUTUAL AGREEMENT & FAMILIAL PROPERTY

We compute the LLM judges' mutual agreement on the instances to explore how "alike" or consistent they are across a set of evaluations. We denote two judges *agree* on an instance if their judgment choices are identical. Then the *mutual agreement* between two judges on a benchmark is defined as the proportion of their agreed instances. Fig. 5(c)(e) displays the mutual agreement heatmap for all judges on MTBench and DevBench, respectively. For MTBench that utilizes the 3-option mode, we also consider the "without tie" agreement since two judges are less disagreed when one chooses {C} while the other prefers a certain solution, compared to the case when they prefer different solutions. The "without tie" agreement heatmap of the twelve judges on MTBench is explored in Fig. 5(d).

The heatmaps reveal clear "familial patterns" in the judgment choices of these LLM judges. For instance, the GPT-4, GPT-4-Turbo, and GPT-4o series exhibit high agreement on MTBench, achieving over 70% with ties included and over 85% without. GPT-3.5-Turbo didn't agree with the GPT-4 series and o1-mini for around 40% of the instances, indicating that they are considerably different in judging capabilities.

For Claude-3 models, similar familial patterns could be observed. Claude-3-Opus highly agrees with Claude-3.5-Sonnet, probably due to their similar capabilities, while it also highly agrees with Claude-3-Sonnet, likely due to their similar model structure within the same series. Interestingly, Claude-3.5-Sonnet and Claude-3-Sonnet do not exhibit a significantly high agreement, indicating that the upgrade from series 3 to 3.5 considerably impacts their judging capabilities.

Gemini models exhibit rather low mutual agreement and "familial property" is minimal, but the most capable Gemini-1.5-pro aligns more closely with other capable models like the GPT-4 series and Claude-3-Opus.

These patterns suggest that familial similarities, possibly stemming from analogous model sizes, training data, and strategies, influence the positional preferences of these judges. In particular, the LLM judges could be primarily grouped by their capabilities; when judging capabilities are comparable, models within the same family series share a higher mutual agreement than across families.

Identifying such groupings provides valuable insights, as comparisons between judges from different groups, both adept at assessing LLM-generated content, can reveal distinct position biases and enrich our understanding of this phenomenon.

## C.2 DISAGREEMENT & BENCHMARK DESIGN INSIGHT

Since the mutual agreement between LLM judges is not perfect and usually a considerable proportion of instances are difficult for them to reach a consensus, disagreement analysis becomes crucial and insightful. Therefore, we define the *disagreement* of an evaluation instance to be the number of judgments different from the majority. By this definition, an instance with all judges reaching a consensus on the better solution will have a disagreement of 0; in contrast, an instance where judgments are widely varied will result in a high disagreement. For our study where twelve judges are investigated, the maximum disagreement of an MTBench is 8, accounting for the 4{A}-4{B}-4{C} choice pattern by 3-option mode. On the other hand, for DevBench instances, the maximum possible disagreement is 6, representing the 6{A}-6{B} judgment distribution for the 2-option mode.

The distributions of instances with different disagreement values on MTBench and DevBench are shown in Fig. 5(a) and Fig. 5(b), respectively. From our disagreement analysis, at least 75% of the judges reached a choice consensus on more than half of the instances on both benchmarks. These are likely easy-to-evaluate instances, and the reliability of LLM judgments is enhanced by majority voting. In comparison, the instances with the highest disagreement are likely the ones that are difficult to evaluate and where the position bias is most likely to occur. However, luckily, these instances are rare, occupying only 0.03% and 0.07% of MTBench and DevBench respectively. In other words, majority voting of multiple capable LLM judges could be practically useful for over 99% of evaluation instances on both benchmarks.

Moreover, if we roughly consider the disagreement value of instances as their difficulty for judging, then Fig. 5(a) and Fig. 5(b) exhibit a balanced distribution of instances with varied difficulty. This is because, except for the instances with the highest disagreement, the numbers of other instances with varied disagreement do not vary significantly, indicating a smoothly increasing difficulty curve across the benchmark datasets.

To summarize, the practical implications of the disagreement analysis are three-fold. First, it helps identify the instances that are difficult or trivial to judge, benefiting benchmark designs to control the difficulty of evaluation by managing the number of these instances across the dataset. Second, it assists in filtering out instances where majority voting of LLM evaluators are likely to offer reliable judgments without direct comparison with human-annotated evaluations, enhancing the scalability of LLM judges especially when human evaluations are costly. In other words, if one-shot judgments from only one LLM judge are not enough reliable, multiple capable LLMs and the majoring voting strategy could be employed to make the evaluation more convincing. Last but not least, disagreement analysis provides a convenient way to make the difficulty variance of instances varied across the dataset tangible. Since the difficulty of an evaluation instance is closely related to the quality gap between the two solutions and hence position bias, the investigation of the instances where most judges particularly disagree with one another could provide more insights and inspiration for future benchmark designs and potential mitigation strategies for position bias.

## D MORE DISCUSSIONS ON ANSWER QUALITY GAP

Intuitively, answer pairs with larger quality gaps are easier for LLM judges to evaluate, as one answer is clearly superior. Therefore, we expect LLM judges to exhibit weaker position bias in such cases. This motivates us to quantify the answer quality gap and investigate its impact on position bias. The quantification utilizes the *overall win rate* ($owr$) of a candidate model (i.e., answer provider) on the other to compute the quality gap. For a set of questions on a certain task and the corresponding solutions by two candidate models, an LLM judge's evaluations would pick out the candidate winner. For the position consistent judgment instances, we can count the number of a candidate's wins and ties, denoted as $C_w$ and $C_t$ respectively. On the other hand, following the methodology introduced by Zheng et al. (2024b), we consider the inconsistent judgment pairs as a *tie* for both candidates, whose counting number is denoted as $C_I$. This follows the intuition that it is difficult to strictly

pick out a better one between two responses with similar quality, hence increasing the likelihood of position bias.

Therefore, considering both cases, we define the **overall win rate** ($owr$) of a candidate's solution over the other as $owr = \frac{1}{n}[C_w + \frac{1}{2}(C_t + C_I)]$, where $n$ is the total number of judgment pairs involving the two candidate solutions in both original and swapped settings. Then the quality gap ($\delta_q$) is calculated as $\delta_q = |owr - 0.5|$, as 0.5 is the exact tie case for the win rate ranging from 0 to 1. We prefer the **overall win rate** over the **consistent win rate** (calculated by $\frac{C_w}{n_C}$, where $n_C$ is the number of position consistent judgment pairs) to quantify $\delta_q$ to mainly address the **data sparsity** issue. If the proportion of position inconsistent judgments becomes high, the available data becomes sparse and $n_C$ becomes small. In the extreme case, where all judgments are inconsistent, $n_C$ is 0 and the consistent win rate is not even computable, making the measurement inaccurate and impractical. In contrast, the overall win rate effectively captures the "comparable quality" scenario by considering "inconsistency-as-tie" and ensuring all data points contribute to the quantification of the difference between answer pairs in quality.

# E    MORE RESULTS OF POSITION BIAS AND ANSWER QUALITY GAP MEASUREMENT

As shown in Fig. 4(a) and (b), considering all judges together, a larger answer quality gap generally leads to better positional consistency and fairness. In this session, we explore whether the discovery is consistent for each individual judge. Same as Section 2.2, we apply the overall win rate to reflect the answer quality gap for visualization.

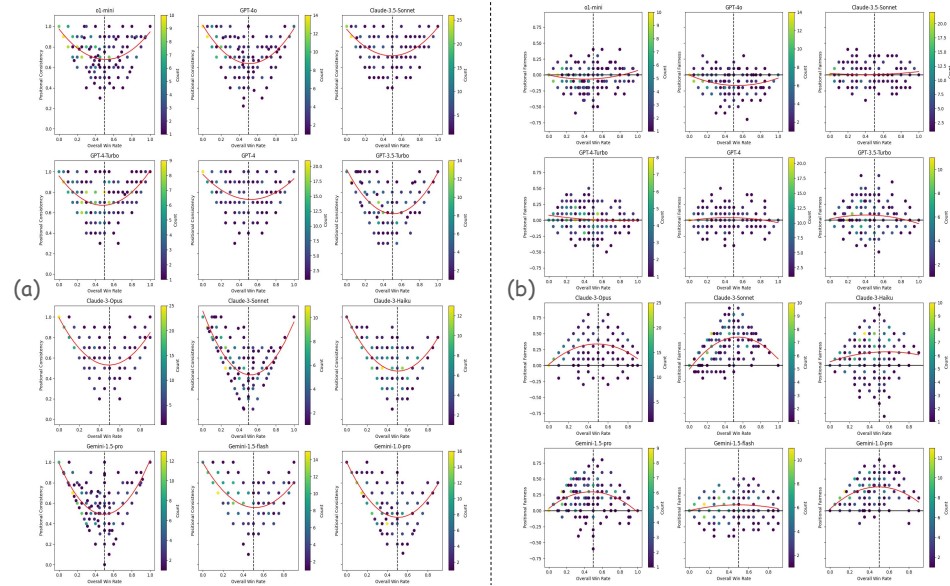

Figure 6: Position Consistency and Preference Fairness vs. overall win rate for each judge on MTBench. Figure (a) refers to the relationship investigation of $PC$ and figure (b) for $PF$.

As shown in Fig. 6 (a), the "parabolic shape" is observed for all individual judges, indicating that the argument "a higher answer quality gap generally results in higher positional consistency" applies to all models. However, Fig. 6 (b) reveals that the positional fairness is more judge-dependent and the impact of the answer quality gap is neglectable for certain judges. For example, while Claude-3-opus and Claude-3-sonnet exhibit conspicuous "parabolic shape", GPT-4 and GPT-3.5 present nearly linear curves. In other words, while the former models align with the general tendency that a larger answer quality gap improves positional fairness, the latter ones preserve fairness regardless of the answer quality gap. This further demonstrates the necessity to investigate positional fairness in addition to consistency when evaluating a judge model's position bias.

# F Variable Selection and Tests

## F.1 Bidirectional Stepwise Regression with AIC

Bidirectional stepwise regression is a combination of forward selection and backward elimination techniques. It iteratively refines the model by adding or removing predictors based on a statistical criterion—commonly the Akaike Information Criterion (AIC). The objective is to select a model that balances goodness of fit and complexity, aiming for the lowest AIC value.

The AIC is given by:

$$\text{AIC} = 2k - 2\log(L), \tag{4}$$

where $L$ is the likelihood of the model and $k$ is the number of parameters in the model, including the error variance $\sigma^2$. For a linear regression model with independent and identically distributed (iid) errors, $N(0, \sigma^2)$, fitted to $n$ observations, the log-likelihood can be written as:

$$\log(L) = -\frac{n}{2}\log(2\pi) - \frac{n}{2}\log(\sigma^2) - \frac{1}{2\sigma^2}\sum_{i=1}^{n}\hat{e}_i^2, \tag{5}$$

where $\hat{e}_i$ is the residual for the $i$th observation, and $\sigma^2$ is the variance of the errors. The AIC, in this context, becomes:

$$\text{AIC} = 2k + n\log(2\pi) + n\log(\sigma^2) + \frac{1}{\sigma^2}\sum_{i=1}^{n}\hat{e}_i^2. \tag{6}$$

This form of the AIC balances the goodness of fit (as reflected by the residual sum of squares) and model complexity (as represented by $k$).

The operation of Bidirectional stepwise regression starts with either no predictors (forward selection) or all predictors (backward elimination), where the model iteratively adds or removes variables. Each step evaluates the impact on the AIC score. In forward selection, variables are added one by one, starting from the null model, such that the addition of each variable results in the largest decrease in AIC. In backward elimination, all variables are included in the model initially, and variables are removed one at a time, with the variable whose removal causes the smallest increase in AIC being dropped.

At each iteration, the change in AIC is computed as $\Delta\text{AIC} = \text{AIC}_{\text{new}} - \text{AIC}_{\text{current}}$, where $\text{AIC}_{\text{new}}$ refers to the AIC after adding or removing a variable, and $\text{AIC}_{\text{current}}$ is the AIC of the current model. If $\Delta\text{AIC} < 0$, the model is improved by the addition or removal of the variable. The process terminates when neither adding nor removing variables results in a lower AIC, signifying that the most parsimonious model, based on AIC, has been reached.

## F.2 Test results

We operated bidirectional stepwise regression on both benchmarks individually and together to identify the factors that are significantly contributing to position bias. Specifically, the variables include lengths (input, output, and prompt), answer quality gap, LLM judges, candidate models, and task categories to predict position consistency and preference fairness respectively. Table 5, 6 records the results of final step in stepwise regression for predicting $PC$ and $PF$, respectively. Table 7, 8 serves for DevBench, and Table 3, 4 is conducted on the integrated set of both benchmarks. The impact of variables on the model is ranked from highest to lowest, from bottom to top. Removed variables listed as None indicate the full model at this given step.

Through benchmark testing, we verified that LLM judges, task categories, and the answer quality gap significantly contribute to position bias in terms of both position consistency and preference fairness. These findings align with our empirical results, showing that position bias varies notably by judge and task, with the answer quality gap being a key influencing factor. The extent of this

impact is reflected by the magnitude of change in AIC when the given variable is removed. It is worth noting that while task output length remains a significant predictor for $PF$ and $PC$ in both benchmarks, the change in AIC magnitude after removing this variable is very minimal. This is consistent across both benchmarks individually and combined. We therefore conclude that, although position bias is influenced by task output length, this dependency is minimal.

Table 3: Final results of stepwise model selection for both benchmarks: Position Consistency

| Removed Variables | DF | Sum of Sq | RSS | AIC |
|---|---|---|---|---|
| None | | | 163.75 | -18370 |
| Task | 20 | 2.832 | 166.59 | -18319 |
| Candidate | 38 | 4.472 | 168.23 | -18303 |
| Quality gap | 1 | 21.953 | 185.71 | -17703 |
| Judge | 13 | 55.417 | 219.17 | -16846 |

Table 4: Final results of stepwise model selection for both benchmarks: Preference Fairness

| Removed Variables | DF | Sum of Sq | RSS | AIC |
|---|---|---|---|---|
| None | | | 254.28 | -16103 |
| Task output length | 1 | 0.836 | 255.12 | -16088 |
| Quality gap | 1 | 11.339 | 265.62 | -15873 |
| Task | 21 | 16.177 | 270.46 | -15817 |
| Judge | 13 | 82.069 | 336.35 | -14641 |

Table 5: Final results of stepwise model selection for MTBench: Position Consistency

| Removed Variables | DF | Sum of Sq | RSS | AIC |
|---|---|---|---|---|
| None | | | 61.974 | -13312 |
| Task output length | 1 | 0.0553 | 62.029 | -13311 |
| Candidate | 29 | 1.6474 | 63.621 | -13282 |
| Task | 7 | 1.5304 | 63.504 | -13244 |
| Judge | 13 | 15.3637 | 77.338 | -12594 |
| Quality gap | 1 | 15.6206 | 77.594 | -12559 |

Table 6: Final results of stepwise model selection for MTBench: Preference Fairness

| Removed Variables | DF | Sum of Sq | RSS | AIC |
|---|---|---|---|---|
| None | | | 129.00 | -10909.2 |
| Quality gap | 1 | 1.931 | 130.93 | -10861.3 |
| Task | 7 | 9.295 | 138.29 | -10689.4 |
| Judge | 13 | 58.847 | 187.85 | -9672.5 |

Table 7: Final results of stepwise model selection for DevBench: Position Consistency

| Removed Variables | DF | Sum of Sq | RSS | AIC |
|---|---|---|---|---|
| None | | | 55.382 | -6940.2 |
| Task output length | 1 | 0.257 | 55.638 | -6933.2 |
| Candidate | 9 | 1.514 | 56.896 | -6905.4 |
| Quality gap | 1 | 13.128 | 68.510 | -6525.3 |
| Judge | 13 | 84.760 | 140.141 | -5146.6 |

Table 8: Final results of stepwise model selection for DevBench: Preference Fairness

| Removed Variables | DF | Sum of Sq | RSS | AIC |
|---|---|---|---|---|
| None | | | 60.104 | -6753.9 |
| Task output length | 1 | 0.061 | 60.165 | -6753.9 |
| Candidate | 9 | 0.731 | 60.834 | -6748.2 |
| Task | 13 | 1.305 | 61.408 | -6737.8 |
| Quality gap | 1 | 1.783 | 61.886 | -6698.6 |
| Judge | 13 | 80.875 | 140.979 | -5108.9 |

## G  DEVBENCH

This session includes a similar baseline comparison analysis on DevBench as on MTBench. As shown in Fig.7, position bias is judge-dependent and task-dependent on DevBench as well, as $PC$ and $PF$ vary significantly across judges and tasks. Similarly, although GPT-4 stands as the baseline model with a generally high $PC$ across tasks, certain models achieve comparable or superior performances on certain tasks. For instance, for *architecture design* evaluations, GPT-4-Turbo, GPT-4o, and Gemini-1.5-pro all surpass GPT-4. Gemini-1.5-pro is especially outstanding, also exceeding GPT-4 in *uml class* evaluations. However, GPT-4 is still the best-performing model on *UML sequence* evaluations, with only GPT-3.5-Turbo can achieve comparable performance regarding certain detailed metrics (e.g., interaction complexity). These discoveries, aligning with the findings on MTBench, further necessitate the need to consider the trade-offs between positional consistency and fairness when selecting the optimal judge model for certain tasks.

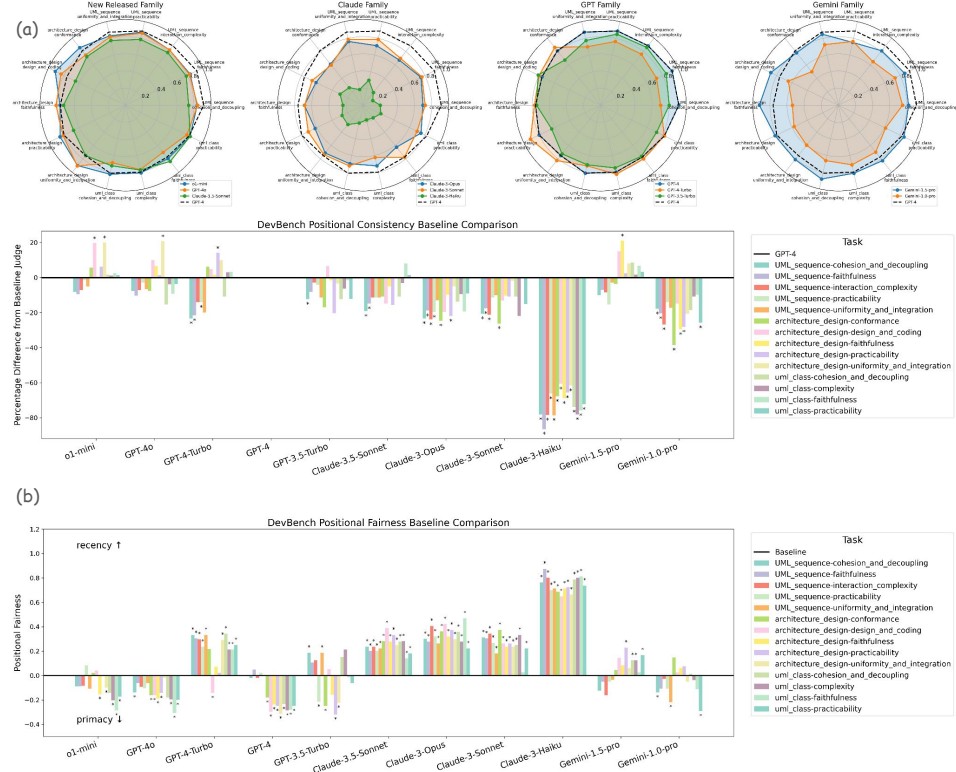

Figure 7: Baseline comparisons of judges across tasks on DevBench. An asterisk marks the statistical significance by Student's t-tests. Figure (a) includes a radar chart comparison by family and an overall baseline bar chart comparison of $PC$. The baseline model is chosen to be GPT-4 for its generally superior performances. Figure (b) demonstrates the $PF$ score of evaluated judges across tasks where the baseline is $PF = 0$.

# H  EXPERIMENT SETTINGS

This session specifies more detailed information about the judges, answer-generating models, tasks, and prompt templates used in this study. We choose to evaluate MTBench and DevBench for the following reasons: (1) all necessary information about the benchmark models, tasks, and questions is publicly available, making modifications convenient (2) they include a wide variety of answer-generating models, tasks, and task questions for a comprehensive evaluation (3) their human evaluations validated the reliability of state-of-the-art judging models (GPT-4 and GPT-4-Turbo) on their evaluation instances, hence model untested by prior work, if reaching high agreement with these validated judges, can be perceived reliable as well.

## H.1  JUDGES, CANDIDATES, AND TASKS

**Judge**  In this study, we choose seven **GPT**, four **Claude**, and three **Gemini** models as the judges. The specific versions for API call are specified as follows: o1-mini-2024-09-12 for **o1-mini**, gpt-4o-2024-05-13 for **GPT-4o**, gpt-4-1106-preview for **GPT-4-Turbo**, gpt-4-0613 for **GPT-4**, and gpt-3.5-turbo-1106 for **GPT-3.5-turbo**; claude-3-5-sonnet-20240620, claude-3-opus-20240229, claude-3-sonnet-20240229, and claude-3-haiku-20240307 for **Claude** series. The other model names and versions are as they are.

**Model**  The reference (or baseline) answer-generating models are **vicuna-13b-v1.3** for MTBench and **human** for DevBench. They are chosen to ensure a baseline quality of responses and an expected widely spread quality gap across evaluations. The other models that are compared to the reference models, namely "Model" in our context, are listed as follows.

- **MTBench (30)**: alpaca-13b, baize-v2-13b, chatglm-6b, claude-instant-v1, claude-v1, dolly-v2-12b, falcon-40b-instruct, fastchat-t5-3b, gpt-3.5-turbo, gpt-4, gpt4all-13b-snoozy, guanaco-33b, guanaco-65b, h2ogpt-oasst-open-llama-13b, koala-13b, llama-13b, mpt-30b-chat, mpt-30b-instruct, mpt-7b-chat, nous-hermes-13b, oasst-sft-4-pythia-12b, oasst-sft-7-llama-30b, palm-2-chat-bison-001, rwkv-4-raven-14b, stablelm-tuned-alpha-7b, tulu-30b, vicuna-33b-v1.3, vicuna-7b-v1.3, wizardlm-13b, wizardlm-30b

- **DevBench (10)**: codellama-7b-instruct, codellama-13b-instruct, codellama-34b-instruct, deepseek-coder-1.3b-instruct, deepseek-coder-6.7b-instruct, deepseek-coder-33b-instruct, gpt-3.5-turbo-1106, gpt-4-0125-preview, gpt-4-0613, gpt-4-1106-preview

The model names are exactly what MTBench Zheng et al. (2024b) and DevBench Li et al. (2024) used in their studies. That is why for GPTs, DevBench specifies the exact version (e.g., gpt-4-0613) while MTBench doesn't (e.g., gpt-4). In this study, we directly use the provided answers of these models to the task questions to form answer pairs and queries for the LLM judges.

**Task**  For tasks, we also follow the original studies of these two benchmarks, except for DevBench we separate the *gerenal* metrics into detailed ones and considered them as different tasks. In this sense, our study experiments on the following tasks to provide a comprehensive study on the positon bias of LLM-as-a-Judge:

- **MTBench (8)**: coding, extraction, humanities, math, reasoning, roleplay, stem, and writing.
- **Devbench (14)**:
  - **UML class** (4): cohesion_and_decoupling, complexity, practicability, and faithfulness
  - **UML sequence** (5): cohesion_and_decoupling, interaction_complexity, practicability, uniformity_and_integration, and faithfulness
  - **architecture design** (5): conformance, design_and_coding, practicability, uniformity_and_integration, and faithfulness

## H.2  PROMPT SETTINGS

We follow the original prompt settings of MTBench and DevBench in our study of pairwise comparative LLM-as-a-Judge.

Though written differently, these prompts all share same key components:

- A system prompt explaining the judging task and the role the LLM should be playing.
- Emphasized "should" and "shouldn't"s.
- A prompt structure with placeholders for specific questions and model answers
- A specified output format for later judgment extraction
- Chain-of-Thought Wei et al. (2022) prompts requiring the LLM judge to provide reasons for its judgment

The detailed prompt templates are specified below.

**For DevBench (evaluating guidance is the evaluating metrics that we seperated, e.g., practicability)**

> Please evaluate the two responses (Response 1, Response 2) based on the provided scoring criteria.
>
> Scoring criteria: <Evaluating Guidance>
> - If the response is incomplete or misses any required key component, regard it as a bad one. - If the response is verbose and/or repetitive, consider it negatively based on the extent. - If the response is well-formatted and clearly-structured, give it extra credit.
>
> **Important**: You should act as an IMPARTIAL judge and be as OBJECTIVE as possible. AVOID ANY POSITION BIASES and ensure that the ORDER in which the responses were presented DOES NOT influence your decision.
>
> Please choose from the following two options based on the scoring criteria:
>
> - A. Response 1 is better than Response 2.
> - B. Response 2 is better than Response 1.
>
> Please provide detailed reasons for your choice. Also, you should pay adequate and the same attention to both responses. Your output should be in the following format:
>
> ```
> Choice: A
> Reason:
> 1. xxxxxx
> 2. xxxxxx
> ......
> ```

**For "math" tasks of MTBench:**

Please act as an impartial judge and evaluate the quality of the responses provided by two AI assistants to the user questions. Your evaluation should consider correctness and helpfulness. You will be given reference answers, the assistant A's answers, the assistant B's answers. Your job is to determine which assistant provides correct and helpful answers to the second user question. Begin your evaluation by comparing both assistants' answers with the reference answers. Identify and correct any mistakes. Avoid any position biases and ensure that the order in which the responses were presented does not influence your decision. Do not allow the length of the responses to influence your evaluation. Do not favor certain names of the assistants. Be as objective as possible. After providing your explanation, output your final verdict by strictly following this format: "[[A]]" if assistant A is better, "[[B]]" if assistant B is better, and "[[C]]" for a tie.

Prompt template:

```
<|The Start of Reference Answer|>
### User:
{question_1}
### Reference answer:
{ref_answer_1}
### User:
{question_2}
### Reference answer:
{ref_answer_2}
<|The End of Reference Answer|>

<|The Start of Assistant A's Conversation with User|>
### User:
{question_1}
### Assistant A:
{answer_a_1}
### User:
{question_2}
### Assistant A:
{answer_a_2}
<|The End of Assistant A's Conversation with User|>

<|The Start of Assistant B's Conversation with User|>
### User:
{question_1}
### Assistant B:
{answer_b_1}
### User:
{question_2}
### Assistant B:
{answer_b_2}
<|The End of Assistant B's Conversation with User|>
```

**For other tasks of MTBench:**

Please act as an impartial judge and evaluate the quality of the responses provided by two AI assistants to the user questions. You should choose the assistant that follows the user's instructions and answers the user's questions better. Your evaluation should consider factors such as the helpfulness, relevance, accuracy, depth, creativity, and level of detail of their responses. You should focus on who provides a better answer to the second user question. Begin your evaluation by comparing the responses of the two assistants and provide a short explanation. Avoid any position biases and ensure that the order in which the responses were presented does not influence your decision. Do not allow the length of the responses to influence your evaluation. Do not favor certain names of the assistants. Be as objective as possible. After providing your explanation, output your final verdict by strictly following this format: "[[A]]" if assistant A is better, "[[B]]" if assistant B is better, and "[[C]]" for a tie.

Prompt template:

```
<|The Start of Assistant A's Conversation with User|>

### User:
{question_1}

### Assistant A:
{answer_a_1}

### User:
{question_2}

### Assistant A:
{answer_a_2}

<|The End of Assistant A's Conversation with User|>

<|The Start of Assistant B's Conversation with User|>

### User:
{question_1}

### Assistant B:
{answer_b_1}

### User:
{question_2}

### Assistant B:
{answer_b_2}

<|The End of Assistant B's Conversation with User|>
```

# I  REPRODUCIBILITY

Our experiments mainly depend on API. Code will be released upon acceptance.

