# OpenReview forum: "Judging the Judges: A Systematic Investigation of Position Bias in Pairwise Comparative Assessments by LLMs"
_ICLR.cc/2025/Conference — ICLR 2025 Conference Withdrawn Submission_

### Official Review · Reviewer_6VD2 · 2024-10-27

**Soundness:** 3
**Presentation:** 3
**Contribution:** 2
**Rating:** 3
**Confidence:** 4

**Summary:**

The work introduces a framework to study bias encountered when using LLMs as effective judges in relation-based pairwise assessments. Given two solutions, LLM judges are prompted to choose the better candidate to the task in question. The authors focus on three aspects of the position bias : Repetition Stability (RS), Position Consistency (PC) and Preference Fairness (PF). Of this, Repetition Stability (RS) evaluates the reliability of the LLM judges when presented with the same queries multiple times. Position Consistency (PC) computes how frequently a LLM judge prioritizes the same candidate when the order of candidates is swapped. Preference Fairness (PF) measures the extent to which judge models favor certain solution positions. They also provide a list of Judge-level, Candidate-level and Task-level factors that affect position bias namely : Familial Property, Answer-quality gap, Task Input/Output Length and Prompt length.

Note : There is a typo in the first word of the title in the paper.

**Strengths:**

1. The work addresses an important issue - effective use of LLMs as judges.
2. The illustrations used by the authors to explain concepts are nice.
3. The experiments seem sane and the claims made in the initial sections of the paper seem to have been verified.

**Weaknesses:**

1. This work is a behavioral exploration of how LLMs perform in judging pairwise relational assessments. In my humble opinion, this type of work does not have much novel ML contribution to be included in conferences like ICLR. It also does not significantly improve our understanding of why LLMs exhibit primacy/recency bias. Perhaps it is better suited as an advanced technical blog for end users or in an NLP conference?

2. The kind of analysis done is quite similar to works that have studied position bias, repetition etc in single and multi-document summarization. The authors could perhaps cite or build upon works like [1,2,3], if they wish.

3. Parts of the paper can be written to make them more intuitive for readers. For example the section explaining Preference Fairness.

[1] Dey, Alvin, et al. "Corpora evaluation and system bias detection in multi-document summarization." arXiv preprint arXiv:2010.01786 (2020).

[2] Kryściński, Wojciech, et al. "Neural text summarization: A critical evaluation." arXiv preprint arXiv:1908.08960 (2019).

[3] Jung, Taehee, et al. "Earlier Isn't Always Better: Sub-aspect Analysis on Corpus and System Biases in Summarization." arXiv preprint arXiv:1908.11723 (2019).

**Questions:**

1. Perhaps it would be useful if some actionable insights were paired with this study. Is there a way to phrase the prompt so that is does not show as much position bias?

2. Do variations in the system prompt affect the result? Does prompting the LLM to make position bias aware decisions in the system prompt help?

3. Shouldn't a fairness metric like Preference Fairness have equal value for all completely Position Incosistent situations?

---

### Official Review · Reviewer_6sPQ · 2024-11-03

**Soundness:** 2
**Presentation:** 3
**Contribution:** 2
**Rating:** 5
**Confidence:** 2

**Summary:**

This paper proposes an evaluation framework for measuring the degree of position bias exhibited in an LLM-as-a-judge scenario for pairwise comparison problems. This framework is used to examine the position bias displayed by a set of LLM-as-a-judge models for two benchmark problems. The results show that for most models position bias is not a significant issue, but that there are variations - some models exhibit more than others. Detailed analysis is presented in a set of appendices and some recommendations for evaluation design are presented.

**Strengths:**

The main strengths of the paper are:

- The paper addresses an interesting niche problem that has not been extensively studied.
- A comprehensive set of evaluations are presented in the appendices included in the paper.
- Some interesting recommendations for other evaluations are presented.

**Weaknesses:**

The main weaknesses of the paper are:

- The statement of the main finding of the paper "Position Bias of Capable Judges are not Mere Random Variations " uses a slightly unusual phrasing. Would it not be better to simply state something like "Capable Judges are not Found to exhibit Strong Position Bias"?  The main message does not come across very clearly.

- The paper would benefit from careful review and revision as at times it is difficult to follow.

- The findings form the paper are interesting, but the amount of actionable advice that emerges from it (Lines 526 - 529) is somewhat limited. Readers may struggle to significantly redesign evaluations based on this. The authors could provide more detailed guidelines or examples of how their findings could be applied to redesign evaluations - perhaps some of the materiel form Appendix A?

- A lot of the content is included in appendices which limits the value of the main paper without these appendices. the authors could better balance the material - for example some of the material from Appendix A could be moved to the conclusions section of the paper and some of the results from Appendix G could be integrated into the main paper.

**Questions:**

As well as addressing the weaknesses mentioned above the authors could consider the following:

- Appendices better referred to as "Appendix D" rather than "Appendix. Sec. D."

- On Line 307 a short introduction to DevBench would really help the reader.

- Text in Fig 1 is very small and hard to read.

- Plots in Figure 3 are too small to read, making them uninformative. Figure 3 also needs better labelling and explanation. E.g. what does the radar plot actually show?

---

### Official Review · Reviewer_RH91 · 2024-11-04

**Soundness:** 2
**Presentation:** 2
**Contribution:** 1
**Rating:** 3
**Confidence:** 4

**Summary:**

The paper studies the problem biases of LLM-as-a-judge in making pairwise evaluations. Specifically, they focus on position bias, where the change in order of options in the prompt, changes the judgement and "preference fairness" where change in option label can yield different judgements. The claimed contributions are in formulating these concepts, identifying factors that impact position bias, and insights for future benchmark design.

In terms of factors, the main presents a taxonomy of factors and how the relate to various levels. In terms of insights for benchmark design, the authors suggest "reducing hard to-evaluate, highly subjective instances" and "minimizing trivial cases". The paper does an empirical analysis of several popular models to support the main findings.

**Strengths:**

The paper addresses an important problem of biases in LLM-as-a-judge style evaluators. While this work focuses on pairwise evaluations, given the broad applicability of LLMs in this manner, this is an area of significance. Some ideas on preference fairness appear to be new (although see the Weakness section).

**Weaknesses:**

Key terms used in the presentation should be defined. What do you mean by "preference fairness" or "primacy preferred"? Why does this preference have "direction"? These concepts should be ideally be formalized and exposition around it improved so that the definitions are clear. Section 2.2. on factors impacting position bias should be improved for clarity. All figures have very small text that are unreadable.

Overall, the contributions are rather limited.

On contribution related to identification of biases: The topics described by this paper has been covered by others. Positional bias has been reported in several works. Previous work at ICLR by Zheng et al. on "Large Language Models Are Not Robust Multiple Choice Selectors" study similar themes are report on the aforementioned biases and ways to mitigate them. Therefore evaluation alone does not merit significance. The two other contributions on are somewhat speculative.

The error rates reported for LLama are surprisingly poor and contradict some of the findings in the literature. A 100% error rate for DevBench suggests something is amiss. Please see papers that use LLama as a baseline, e.g. "Aligning with Human Judgement: The Role of Pairwise Preference in Large Language Model Evaluators" and "Fairer Preferences Elicit Improved Human-Aligned LLM Judgments" which report on other datasets. I would encourage a recheck of results here.

**Questions:**

NA

---

### Official Review · Reviewer_1rdV · 2024-11-09

**Soundness:** 3
**Presentation:** 3
**Contribution:** 2
**Rating:** 5
**Confidence:** 4

**Summary:**

This paper proposes a systematic framework for examining the position bias of LLM-based judges. Using this framework, the authors analyze the position bias across various state-of-the-art LLMs on two benchmarks: MTBench and DevBench. In doing so, the authors find that position bias varies by judge/task, and that it is strongly correlated with the gap in quality between answers and weakly correlated with average task output length.

**Strengths:**

- The evaluation framework, particularly decoupling the repetition stability and preference fairness metrics from position consistency, provides a robust and consistent framework for future research on the reliability of LLM-based judges.
- The empirical results are comprehensive, covering 12 judge models across more than 100,000 instances.
- Based on the experimental results, the authors draw reasonable and intuitive insights (e.g., LLM-based judges are more likely to exhibit position bias when the candidate answers are of similar quality).

**Weaknesses:**

- It’s quite strange to consider the position bias of LLM-based judges in isolation of their accuracy (e.g., the rate at which the judge selects the objectively better answer, or at least the consensus answer based on crowdsourcing). In doing so, the authors tend to over-generalize claims based on the characteristics of the LLM-based judges with respect to position bias (e.g., that GPT-3.5-Turbo “may be employed as a cost-effective alternative to coding evaluations” since it achieves a high preference fairness score and a comparable position consistency metric to GPT-4 and GPT-4o) that are likely unfounded given a more holistic view of LLM-based judges.

- Seemingly the key insight drawn is that LLM-based judges tend to exhibit position bias when the answer quality gap is small, which is interesting but not surprising, nor is very actionable. It would have been nice had the authors studied some more actionable factors related to the design of the judge such as prompt template, option modes, etc.

**Questions:**

- Regarding measuring preference fairness, why are normalized primacy/recency-count-numbers overly sensitive to dataset size? Can you provide an intuitive example of this? Do the normalized metrics for repetition stability and position consistency have similar issues? How are $S_{min}^{-}$ and $S_{max}^{+}$ computed?
- Are these three metrics (repetition stability, position consistency, preference fairness) comparable across datasets? If so, is there a significant difference in the frequency with which LLM-based judges exhibit position bias between MTBench and DevBench? If so, can you draw any conclusions as to which factors contribute to the difference?
- How does the computation of the answer quality gap change in a two-mode setting (e.g., no ties like on DevBench)? Is $C_t$ just 0? Are there other important considerations for a two-mode vs three-mode setting, e.g., presumably it’s easier to have higher repetition stability in the two-mode setting vs. the three-mode setting since there are less options?
- Can you provide insight into why the Gemini-1.5-flash exhibits a near 1 error rate on DevBench, but an error rate of 0 on MTBench?

- Other comments:
Sec. 3.2 line 353: “arean -> arena”.
Sec. 4 line 374: “Table 4 -> Table 2”.
Sec. 4 fig. 3b is seemingly missing.
Sec. 4 line 473: “Fig. 1(b)” -> “Fig. 4(b)”

---

### Note · Authors · 2024-11-26

I have read and agree with the venue's withdrawal policy on behalf of myself and my co-authors.